# Keeping university open did not increase the risk of SARS-CoV-2 acquisition: A test negative case-control study among students

**Erika Renzi**[1]*, **Valentina Baccolini**[1], **Antonio Covelli**[1], **Leonardo Maria Siena**[1], **Antonio Sciurti**[1], **Giuseppe Migliara**[2], **Azzurra Massimi**[1], **Carolina Marzuillo**[1], **Corrado De Vito**[1], **Leandro Casini**[3], **Antonio Angeloni**[4], **Ombretta Turriziani**[5,6], **Guido Antonelli**[5,6], **Fabrizio D'Alba**[7], **Antonella Polimeni**[8], **Collaborating Group**[¶], **Paolo Villari**[1]

1 Department of Public Health and Infectious Diseases, Sapienza University of Rome, Rome, Italy,
2 Department of Life Sciences, Health and Health Professions, Link Campus University, Rome, Italy,
3 Special Office for Prevention, Protection and H&S Overall Supervisory, Sapienza University of Rome, Rome, Italy, 4 Department of Experimental Medicine, Sapienza University of Rome, Rome, Italy,
5 Department of Molecular Medicine, Sapienza University of Rome, Rome, Italy, 6 Microbiology and Virology Unit of the University Hospital "Policlinico Umberto I", Rome, Italy, 7 University Hospital "Policlinico Umberto I", Rome, Italy, 8 Department of Oral and Maxillofacial Science, Sapienza University of Rome, Rome, Italy

¶ Membership of the Collaborating Group is provided in the Acknowledgments.
* erika.renzi@uniroma1.it

**Data Availability Statement:** All relevant data are within the manuscript and its Supporting information files.

## Abstract

### Background

During the SARS-CoV-2 testing program offered through the RT-PCR test by Sapienza University of Rome, we conducted a test-negative case-control study to identify risk factors for acquiring SARS-CoV-2 infection among university students.

### Methods

Each SARS-CoV-2-positive case detected was matched to two controls randomly selected from students who tested negative on the same day. 122 positive students and 244 negative students were enrolled in the study. Multivariable conditional logistic regression models were built. Adjusted odds ratios (aORs) and 95% confidence intervals (CIs) were calculated. A second model was limited to students who had attended campus.

### Results

Out of 8223 tests for SARS-CoV-2, 173 students tested positive (2.1%), of whom 122 (71.5%) were included in the case-control study. In the first analysis, being a non-Italian student (aOR: 8.93, 95% CI: 2.71–29.41), having received only the primary vaccination course (aOR: 2.94, 95% CI: 1.24–6.96) compared to the booster dose, known exposure to a COVID-19 case or someone with signs/symptoms suggestive of COVID-19 (aOR: 6.51, 95% CI: 3.48–12.18), and visiting discos (aOR: 4.07, 95% CI: 1.52–10.90) in the two weeks before testing increased the likelihood of SARS-CoV-2 infection. Conversely, students attending in-person lectures on campus seemed less likely to become infected (aOR: 0.34,

**Funding:** The author(s) received no specific funding for this work.

**Competing interests:** The authors declare that they have no competing interests.

95% CI: 0.15–0.77). No association was found with other variables. The results of the second model were comparable to the first analysis.

## Conclusions

This study indicates that if universities adopt strict prevention measures, it is safe for students to attend, even in the case of an infectious disease epidemic.

## Introduction

The transmission of SARS-CoV-2 is dependent on the normal interactions of daily life and therefore, during the COVID-19 pandemic, most countries opted to reduce economic, educational and recreational activities to disrupt the spread of the virus; however, this had considerable public health, economic and social impacts [1]. In this context, the identification of risk factors for the transmission of SARS-CoV-2 infection is crucial to guide health policy decisions aimed at limiting virus transmission and, at the same time, preserving the normal activities of everyday life as far as possible [2, 3]. However, few analytical epidemiological studies have been conducted worldwide to investigate the risk factors for SARS-CoV-2 acquisition [4–13], particularly in healthy individuals attending community settings and areas posing unique challenges for transmission control, such as schools [14–16] and higher education institutions (HEIs) [17–20].

Immediately after the start of the pandemic, in many countries schools and HEIs replaced face-to-face lectures with remote teaching activities [21, 22]. The prolonged closure of HEIs led to a worsening in students' performance, mental health and well-being, suggesting that interruption of face-to-face teaching activities is not a sustainable long-term measure [23–25]. Subsequently, face-to-face teaching activities were partially resumed, albeit with the requirement for strict preventive measures. These included a reduction in classroom capacity, face-to-face/distance hybrid classes, enforced social distancing, handwashing, mask wearing, and (self-)monitoring of symptoms, as well as implementation of the vaccination campaign [21, 26, 27]. The reopening of schools and universities for face-to-face activities was debated at length, given the high risk of infection when individuals are confined to enclosed spaces for long periods of time and the marked impact of distance learning on student performance, socialization and emotional well-being [28]. Nevertheless, the question of whether schools and universities should remain open during a pandemic is still unresolved [29, 30].

For these reasons, identifying the risk factors involved in the transmission of SARS-CoV-2 in HEIs, such as universities whose health policies include the implementation of preventive measures, could provide valuable insights into the role of university closure and reopening in the transmission of COVID-19 in communities, especially after widespread vaccination. Therefore, as part of the SARS-CoV-2 infection testing program for all students of Sapienza University of Rome, a case-control study was conducted after the release of the anti-COVID-19 vaccines on a sample of university students to identify risk factors for SARS-CoV-2 virus acquisition in HEIs.

## Materials and methods

### Setting and participants

A test-negative case-control study was conducted between 8[th] September 2021 and 3[rd] February 2022 during the testing program organized by Sapienza University of Rome, which offered a free RT-PCR (reverse transcriptase polymerase chain reaction) molecular test to all students

enrolled in degree programs (2021–2022 academic year) [19]. Students who confirmed positive for SARS-CoV-2 during the testing program were enrolled as cases. Each case detected was matched to two controls randomly selected from students who tested negative on the same day as the positive case. The control selection process was carried out using free software (https://it.piliapp.com/random/number/) to generate a random sequence of numbers. The control selection procedure identified ten potential controls for each case, who were then contacted in the order of selection until two students agreed to participate in the study. Both cases and controls were first contacted by email in which we explained the study and invited them to give their consent to be contacted by the research staff of the Department of Public Health and Infectious Diseases. Those who agreed to take part in the study underwent structured 15-minute phone interviews carried out in Italian or English within 72 hours of receipt of the test results.

## Data collection

The questionnaire consisted of 39 questions grouped into three sections. The first section examined the demographic and general characteristics of the sample: age, gender, nationality, area of study, year of study, job (if applicable), comorbidities/chronic conditions, and living with a person with a chronic condition.

In the second section we investigated behavior with respect to COVID-19, non-pharmacological preventive measures, and vaccination. Specifically, we asked participants whether they had had a previous SARS-CoV-2 infection, their COVID-19 vaccination status (booster dose, primary vaccination course, primary vaccination course not completed, unvaccinated), and whether any reduction in adherence to COVID-19 non-pharmacological preventive measures (wearing a mask indoors, social distancing, hand washing) had occurred after the release of the COVID-19 vaccines and/or the mandatory possession of an EU digital COVID certificate. We also investigated whether they had had a known exposure to someone with COVID-19 or with signs/symptoms suggestive of COVID-19, defined as being within two meters for a total of ≥15 min without wearing any type of mask within 14 days [31].

The third section explored potential exposures that had occurred in the two weeks prior to the swab. Participants were asked to rate on a five-point Likert-type scale from "never" to "more than once a day" or "always" how often, on average, they had attended lectures and other activities on the university campus (e.g., library, internship, laboratory); they had visited bars or restaurants on or off campus (e.g., for breakfast, lunch, aperitif, dinner or after dinner); they had visited cinemas, theaters, museums, discos, clubs or churches; they had visited a salon or beauty salons, shopping malls or grocery stores; they had guests or had been guests, or had participated in private social or religious gatherings (e.g., parties, ceremonies); they had participated in indoor sports activities (e.g. gym, swimming pool); and they had visited health facilities (general practitioner, hospital). We also asked them whether they had used public transport for either short (within the city) or long distances.

The study was performed in accordance with the World Medical Association Declaration of Helsinki. Participants were asked to sign a written informed consent and the anonymity of the information collected was guaranteed. The institutional ethics board of the Umberto I Teaching Hospital/Sapienza University of Rome approved this study (protocol n. 188/2021). This study followed the Strengthening the Reporting of Observational Studies in Epidemiology (STROBE) reporting guideline [32] (S1 Checklist).

## Statistical analysis

Data on national detection rates of confirmed SARS-CoV-2 infections were collected from the Italian Civil Protection/Ministry of Health website [33]. Descriptive statistics were obtained

using median and interquartile range or mean and standard deviation for continuous variables, and proportions for dichotomous and categorical variables. For the purposes of statistical analysis, students were considered to be Italian or non-Italian; area of study was categorized into healthcare (e.g., medicine, nursing) and non-healthcare; year of study was collapsed into first vs. second year or above; self-reported reduction in adherence to handwashing procedures, maintenance of physical distancing (at least one meter) and face-mask wearing were categorized into two modalities (reduced vs. unchanged); and exposure activity responses during the two weeks before testing for SARS-CoV-2, collected using Likert scales, were dichotomized as never vs. once or more.

Each variable was first examined by univariable conditional logistic regression analysis. Then, a multivariable conditional logistic regression model was built to identify predictors of SARS-CoV-2 infection. Variables were included in the model based on expert opinion. Specifically, potential exposures in the two weeks before testing were adjusted for sex, age, nationality, area of study, COVID-19 vaccination status, reduction in mask use indoors, having a job, and known exposure to a COVID-19 case or someone with sign/symptoms suggestive of COVID-19. Additionally, we collapsed the potential exposures into five categories: activities in the community (both essential and non-essential activities), attending lectures on the university campus, attending other activities on campus (e.g., library, laboratory, internship), visiting bars or restaurants (on or off campus), visiting discos or clubs, and use of public transport (for short or long distances). As a result, the final model consisted of the following variables: age (continuous), gender (dichotomous), nationality (dichotomous), area of study (dichotomous), COVID-19 vaccination status (categorical), having a job (dichotomous), known exposure to a COVID-19 case or someone with signs/symptoms suggestive of COVID-19 (dichotomous), reduction in mask use indoors (dichotomous), activities in the community (dichotomous), attending lectures on campus (dichotomous) and other activities on campus (dichotomous), visiting discos or clubs (dichotomous) and bars or restaurants (dichotomous), and use of public transport (dichotomous). Adjusted odds ratios (aORs) and 95% confidence intervals (CIs) were calculated. A second conditional logistic regression model was restricted to participants who reported attending the university campus in the two weeks before testing (60 cases matched to 88 controls in a 1:1 or 1:2 ratio). The same variables used in the first analysis were considered.

All calculations were performed using Stata (StataCorp LLC, 4905 Lakeway Drive, College Station, TX 322, USA), version 17.0. A two-sided p-value < 0.05 was considered statistically significant.

## Results

### General characteristics of the sample

During the 20-week (8th September 2021–3rd February 2022) testing program, a total of 8223 SARS-CoV-2 RT-PCR tests were administered (weekly mean: 436.5; daily mean: 116.4; daily range: 27–298) (Fig 1A); of these, 173 tests (2.1%) returned positive. Overall, the trend of our weekly detection rate was comparable to that at national level, with the highest proportion of cases recorded in the last seven weeks (December 2021–January 2022) of the campaign, which corresponded to the fourth wave of COVID-19 in Italy (Fig 1B and 1C). Regarding the socio-demographic characteristics of the students who participated in the Sapienza testing program, they were mostly women (68.9%), Italian (97.6%), enrolled in a healthcare degree course (45.3%) or a non-science course (40.8%), and attended the first or second year of study (53.8%). Mean age was 24.4 (± 4.2) years (Table 1).

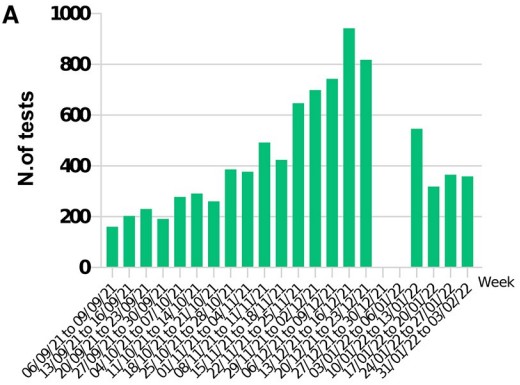

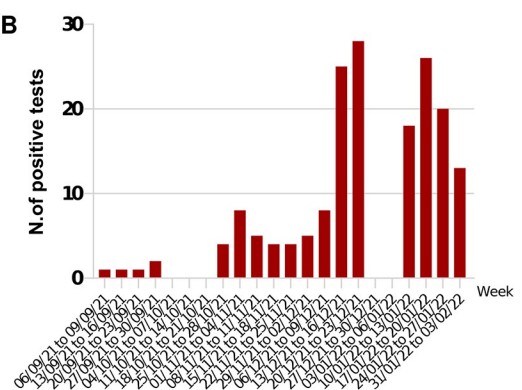

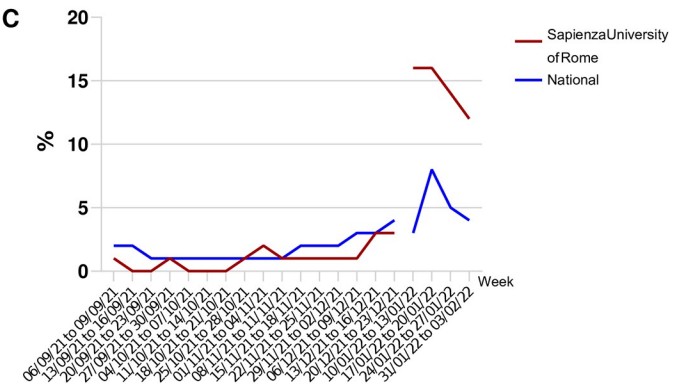

**Fig 1.** Sapienza University testing program, 8th September 2021–3rd February 2022: A) Number of SARS-CoV-2 RT-PCR tests administered; B) number of positive tests detected; C) Weekly detection rates of SARS-CoV-2 RT-PCR positive tests registered through the Sapienza University testing program in comparison to the weekly detection rate at a national level.

Out of 173 positive cases detected during the testing program, a total of 122 cases were enrolled in our case-control study (response rate: 71.5%). Of the remaining 51 cases, 18 students refused to participate to the study, 26 students did not give their consent to be contacted by telephone, and seven did not speak Italian or English fluently. Similarly, 16 potential controls declined to be enrolled, accounting for a control response rate of 93.5%. In total, 366 individuals were interviewed (122 cases, 244 controls).

The general characteristics of the case-control study population are reported in Table 2. Cases and controls were mostly female (73.4% and 72.9%, respectively), with a similar mean age (23.4 ± 3.2 years for cases; 23.2 ± 3.8 years for controls). There were more non-Italian nationals among the cases than the controls (13.9% vs. 2.4%). Most students in both groups attended a healthcare degree course (61.5% and 54.5%, respectively), while a small difference was found in the proportion of students who reported having a job (12.3% of cases vs. 16.0% of controls). Regarding COVID-19 vaccination, controls had received the booster dose more often than cases (47.1% vs. 30.3%), although a similar proportion of students in both groups had not received COVID-19 vaccination or had not completed the primary vaccination course (9.8% vs. 7.0%). Approximately one quarter of the participants had a chronic condition or lived with someone with a chronic condition (23.8% vs. 23.4%). A higher percentage of cases than controls reported a reduction in mask use indoors after the release of the COVID-19 vaccine and/or the

**Table 1. Characteristics of students who were tested at least once for SARS-CoV-2.** Results are expressed as mean (standard deviation, SD), median (interquartile range, IQR), or frequency (percentage).

|  | N | (%) |
|---|---|---|
| Gender |  |  |
| Female | 5665 | (68.9) |
| Male | 2558 | (31.1) |
| Age, years |  |  |
| Mean (SD) | 24.4 | (22–26) |
| Median (IQR) | 24 | (97.6) |
| Country of residence |  |  |
| Italy | 7868 | (45.3) |
| Other | 355 | (13.9) |
| Area of study |  |  |
| Healthcare | 3724 | (68.9) |
| Science | 1140 | (31.1) |
| Other | 3359 | (4.2) |
| Year of study |  |  |
| First | 1844 | (22.4) |
| Second | 2581 | (31.4) |
| Third | 1869 | (22.7) |
| Fourth | 382 | (4.6) |
| Fifth | 363 | (4.4) |
| Sixth | 375 | (4.6) |
| Master degree, doctorate degree, specialization school | 783 | (9.5) |
| Outside prescribed course | 26 | (0.3) |

requirement to possess an EU digital COVID certificate (24.6% vs. 15.2%). Almost half of the cases reported having been in contact with a COVID-19 case or someone with signs and/or symptoms suggestive of COVID-19, compared to a lower proportion of controls (18.0%).

## Exposures in the 14 days before testing for SARS-CoV-2

Regarding potential exposures in the two weeks before testing, cases and controls took part in each investigated activity to a similar extent, with some exceptions (Table 3). The univariable conditional logistic regression model reported a statistically significant difference in the control group regarding in-person lecture attendance on the university campus compared to the case group (47.2% vs 32.0%; p = 0.002). Similarly, cases attended other activities on campus (library, laboratory, internship) less often than controls, although this did not reach statistical significance (39.3% vs 50.0%; p = 0.054). Similarly, controls reported visiting bars or restaurants on or off campus more often than cases (79.5% vs. 68.9%; p = 0.021). Conversely, cases went to discos or clubs more often than controls (18.0% vs. 8.2%, p = 0.005), whereas no meaningful difference was observed for visiting cinemas, theaters, museums or churches; visiting personal grooming salons or beauty salons, shopping centers or grocery stores; attending parties or ceremonies; taking part in indoor sport activities; using public transport for either short or long distances; or attending healthcare facilities.

## Predictors of SARS-CoV-2 infection

In the multivariable conditional logistic regression model, being a non-Italian student (aOR: 8.93, 95% CI: 2.71–29.41), having received only the primary vaccination course (aOR: 2.94,

**Table 2. Student sociodemographic characteristics, vaccination status and self-reported adherence to precautionary measures after vaccine release.** Results are expressed as mean (standard deviation, SD), median (interquartile range, IQR), or frequency (percentage).

| | Cases | Controls | Unadjusted OR (95% CI)* | p-Value* |
|---|---|---|---|---|
| Gender | | | | |
| Male | 33 (26.6) | 65 (27.1) | Ref. | |
| Female | 89 (73.4) | 179 (72.9) | 0.99 (0.60–1.60) | 0.934 |
| Age, years | | | 1.01 (0.95–1.07) | 0.782 |
| Mean (SD) | 23.4 (3.2) | 23.2 (3.8) | | |
| Median (IQR) | 23 (21–24) | 23 (21–24) | | |
| Nationality | | | | |
| Italian | 105 (86.1) | 238 (97.6) | Ref. | |
| Non-Italian | 17 (13.9) | 6 (2.4) | 5.67 (2.23–14.28) | <0.001 |
| Area of study | | | | |
| Healthcare | 75 (61.5) | 133 (54.5) | Ref. | |
| Other | 47 (38.5) | 111 (45.5) | 0.78 (0.51–1.18) | 0.205 |
| Year of study | | | | |
| First | 22 (18.0) | 42 (17.2) | Ref. | |
| Second or above | 100 (82.0) | 202 (82.8) | 1.13 (0.70–1.83) | 0.623 |
| Vaccination status | | | | |
| Booster dose | 37 (30.3) | 115 (47.1) | Ref. | |
| Primary vaccination course | 73 (45.9) | 112 (59.8) | 3.39 (1.75–6.55) | <0.001 |
| Unvaccinated or vaccinated with one dose (primary vaccination course not completed) | 12 (9.8) | 17 (7.0) | 3.46 (1.35–8.84) | 0.010 |
| Having a job | 15 (12.3) | 39 (16.0) | 0.74 (0.39–1.40) | 0.355 |
| Having a chronic condition or living with someone with a chronic condition | 29 (23.8) | 57 (23.4) | 1.02 (0.61–1.73) | 0.929 |
| Reduction in mask use indoors after vaccination | 30 (24.6) | 37 (15.2) | 2.06 (1.12–3.78) | 0.020 |
| Reduction in social distancing after vaccination | 31 (25.4) | 58 (23.8) | 1.10 (0.66–1.82) | 0.726 |
| Reduction in the frequency of hand washing after vaccination | 11 (9.0) | 13 (5.3) | 1.74 (0.76–3.96) | 0.190 |
| Known exposure to a COVID-19 case or someone with signs/symptoms suggestive of COVID-19 ≤ 14 days before SARS-CoV-2 test | 55 (45.0) | 44 (18.0) | 3.79 (2.28–6.32) | <0.001 |

COVID-19: coronavirus disease 2019. OR: odds ratio. CI: confidence interval.

* Univariable conditional logistic regression model for factors associated with SARS-CoV-2 infection.

95% CI: 1.24–6.96) compared to the booster dose, a known exposure to a COVID-19 case or someone with signs/symptoms suggestive of COVID-19 (aOR: 6.51, 95% CI: 3.48–12.18), and visiting discos or clubs (aOR: 4.07, 95% CI: 1.52–10.90) in the two weeks before testing increased the likelihood of SARS-CoV-2 infection. Conversely, students attending in-person lectures on the university campus seemed less likely to become infected (aOR: 0.34, 95% CI: 0.15–0.77). The analysis showed that age, gender, area of study, having a job, reducing mask use indoors after vaccination, attending activities in the community, eating at bar or restaurants, and use of public transport were not predictors of SARS-CoV-2 infection (Table 4, Model 1).

The results of the second model, which was limited to students who reported attending the university campus in the two weeks before testing, were comparable to the first analysis (Table 4, Model 2). Specifically, having received only the primary vaccination course (aOR: 2.01, 95% CI: 0.21–19.36) compared to the booster dose, a known exposure to a COVID-19

**Table 3. Student activity-related exposures ≤ 14 days before testing for SARS-CoV-2.** Results are expressed as frequency (percentage).

| | Cases | Controls | Unadjusted OR (95% CI)* | p-Value* |
|---|---|---|---|---|
| Attend lectures on the university campus | 39 (32.0) | 115 (47.2) | 0.44 (0.27–0.64) | 0.002 |
| Other activities on campus (library, laboratory, internship) | 48 (39.3) | 122 (50.0) | 0.64 (0.41–1.01) | 0.054 |
| Bars or restaurants | 84 (68.9) | 194 (79.5) | 0.54 (0.32–0.91) | 0.021 |
| Cinemas, theaters, museums, churches | 35 (28.7) | 78 (32.0) | 0.85 (0.52–1.38) | 0.511 |
| Discos or clubs | 22 (18.0) | 20 (8.2) | 2.74 (1.35–1.55) | 0.005 |
| Salons/beauty salons, shopping centers, grocery stores, banks, post offices | 99 (81.5) | 214 (87.7) | 0.57 (0.31–1.07) | 0.081 |
| Parties or ceremonies | 68 (55.7) | 135 (55.3) | 1.02 (0.63–1.66) | 0.934 |
| Indoor sporting activities | 21 (17.2) | 61 (25.0) | 0.60 (0.34–1.07) | 0.083 |
| Use of public transport for short distances (bus, metro, car sharing) | 77 (63.1) | 155 (63.5) | 0.98 (0.60–1.60) | 0.933 |
| Use of public transport for long distances (airplane, boat, interregional/international train or buses) | 26 (21.3) | 63 (25.8) | 0.76 (0.45–1.31) | 0.328 |

OR: odds ratio. CI: confidence interval.

* Univariable conditional logistic regression model for factors associated with SARS-CoV-2 infection.

case or someone with signs/symptoms suggestive of COVID-19 (aOR: 3.84, 95% CI: 1.38–10.71) and visiting discos or clubs (aOR: 3.89, 95% CI: 1.02–14.86) were predictors of becoming infected with SARS-CoV-2. No other variables showed a significant association with the outcome.

## Discussion

The operation of schools and universities during a pandemic is obviously a strategic issue. School closures were a common means of controlling the spread of SARS-CoV-2 and the question of how to keep schools open safely has been controversial. One line of evidence supports the claim that schools can be an important SARS-CoV-2 transmission source [30, 34], outlining the methodological limitations of research minimizing SARS-CoV-2 transmission in school, showing, for example, similar infection rates in schools and surrounding communities. Conversely, a systematic review found substantial heterogeneity among school closure studies: half of the studies with a low risk of bias reported a reduction in community transmission of up to 60%, with the remaining half reporting null findings. The majority of the few school reopening studies with a low risk of bias reported no associated increases in transmission [29]. Collectively, the scientific evidence on primary and secondary school closures and reopenings, although still an unresolved issue, agrees that (i) school closures in the early phase of the pandemic were helpful in counteracting the spread of the virus at a time when our understanding of SARS-CoV-2 infection was limited [35]; (ii) the variability in the results of published studies may reflect problems in study design [29, 30, 36]; and (iii) in-person learning increases children's performance and well-being and can be safely maintained in school with robust preventive measures [5, 14, 15, 36, 37]. In contrast, there are fewer studies on the magnitude of universities' contribution to community transmission [37], particularly in respect of strategies that might mitigate the spread of the virus, and the potential benefits of in-person education models on academic, social, mental and physical health outcomes. The prolonged closure of universities has brought far-reaching changes in multiple facets of the student experience, including a decline in academic performance [38, 39], financial issues [40, 41], increased levels of psychological distress and anxiety disorders [42–44], and inequalities in learning opportunities, which particularly affect international students and those with specific learning disorders [39, 45].

**Table 4. Multivariable conditional logistic regression model for SARS-CoV-2 infection among Sapienza University students (Model 1) or restricted to those students that reported attending the university campus in the two weeks before testing (Model 2).**

| | Model 1 | | Model 2 | |
|---|---|---|---|---|
| | aOR (95% CI) | *p*-Value | aOR (95% CI) | *p*-Value |
| Age (years) | 0.99 (0.89–1.08) | 0.747 | 0.99 (0.85–1.59) | 0.943 |
| Gender (female) | 1.52 (0.75–3.10) | 0.244 | 0.66 (0.22–1.97) | 0.453 |
| Nationality (non-Italian) | 8.93 (2.71–29.41) | <0.001 | 4.85 (0.84–27.87) | 0.077 |
| Area of study (non-healthcare) | 1.19 (0.66–2.14) | 0.567 | 1.08 (0.45–2.59) | 0.856 |
| Vaccination status | | | | |
| Booster dose | Ref. | | Ref. | |
| Primary vaccination course | 2.94 (1.24–6.96) | 0.014 | 7.08 (1.54–32.64) | 0.012 |
| Unvaccinated or vaccinated with one dose (primary vaccination course not completed) | 1.97 (0.58–6.69) | 0.277 | 2.01 (0.21–19.36) | 0.546 |
| Having a job (yes) | 0.63 (0.26–1.57) | 0.325 | 0.38 (0.07–2.20) | 0.280 |
| Known exposure to a COVID-19 case or someone with signs/symptoms suggestive of COVID-19 (yes) | 6.51 (3.48–12.18) | <0.001 | 3.84 (1.38–10.71) | 0.010 |
| Reduction in mask use indoors after vaccination (yes) | 1.62 (0.69–3.81) | 0.271 | 1.59 (0.52–4.86) | 0.410 |
| Activities in the community (yes) | 0.52 (0.16–1.72) | 0.286 | 0.26 (0.02–3.47) | 0.308 |
| Attend lectures on the university campus (yes) | 0.34 (0.15–0.77) | 0.010 | — | — |
| Other activities on campus (library, laboratory, stage) (yes) | 0.87 (0.47–1.61) | 0.663 | — | — |
| Discos or clubs (yes) | 4.07 (1.52–10.90) | 0.005 | 3.89 (1.02–14.86) | 0.047 |
| Bars or restaurants (yes) | 0.69 (0.33–1.14) | 0.307 | 0.73 (0.22–2.41) | 0.604 |
| Use of public transport (yes) | 0.85 (0.43–1.67) | 0.641 | 0.66 (0.23–1.91) | 0.445 |

aOR: adjusted odds ratio. CI: confidence interval.

The current study had a rigorous analytical design and compared the demographic characteristics, in-person learning activities within the university and other possible exposures to the virus outside the university in students testing positive or negative for SARS-CoV-2 infection. The study was carried out during Sapienza University's SARS-CoV-2 testing program and took place during the fourth wave of the pandemic in Italy, which was characterized by the spread of the Omicron variant. Our results clearly indicate that students who attended lectures on the university campus were less likely to become infected. This findings, in line with other previous analytic studies performed at Sapienza during the second and third waves of the pandemic in Italy [17, 19], could be explained by the numerous preventive measures implemented within the university to encourage safe reopening: 50% reduction in classroom capacity; hybrid face-to-face/at-distance lectures; low occupancy of all indoor spaces (libraries, laboratories, refreshment areas) and enforcement of face-mask wearing within all campus premises. A widespread campaign was enacted by Sapienza University to encourage students to focus on a combined four-part strategy: washing hands, staying at home in case of signs or symptoms suggestive of COVID-19, physical distancing, and continuous mask use [46]. After the release of the anti-COVID-19 vaccine, the campaign was modified to include active promotion of vaccination adherence by students and university staff [47]. Additionally, a free SARS-CoV-2 testing program for all asymptomatic students and a surveillance system for SARS-CoV-2 cases among students and university staff were instituted during the early stages of the COVID-19 response [17–19], to facilitate the test-trace-isolate-quarantine strategy, activities that may have been particularly useful in contrasting the virus spread in those scenarios with a high virus circulation such as the winter of 2022, during which we registered the highest number of positive cases. Overall, these measures seem contributed to maintaining a safe environment on the campus and may have induced a less risky behavior also outside the university. However,

because the testing program was voluntary, we cannot rule out the possibility that students who attended the university campus were more likely to be tested for screening purposes (i.e., with low or no probability of COVID-19) than non-attending students. On the other hand, this bias could be counteracted if individuals who were tested were overly cautious or if exposed individuals avoided testing because they did not want to have to self-isolate [48]. However, although with a limited sample size, a second analysis in which non-attending students were excluded from the model, so that all study participants had the same exposure conditions, gave findings that did not change significantly in relation to the other variables.

Other results of this study are consistent with the existing scientific literature. As expected, students with confirmed SARS-CoV-2 infection were much more likely than those without the virus to report close contact with a COVID-19 case or an individual with influenza-like illness. Moreover, students who had received only the primary course of COVID-19 vaccine were more at risk of infection with SARS-CoV-2 than those who had received their booster dose. The efficacy of two doses of COVID-19 vaccine in preventing infection and symptomatic disease decreases by 20–30% six months after vaccination [49] and the booster dose, as described in recent studies, seems to increase protection against the Omicron variant [50].

Visiting discos or clubs was associated with an increased risk of SARS-CoV-2 infection, consistent with a large body of evidence on the role of parties in the youth population [5, 8, 12, 18, 20, 50] and the fact that in discos or clubs the recommended controls and preventive measures are more likely to be sidelined [51]. Conversely, social activities often considered high-risk, such as frequenting restaurants, bars or pubs, did not have an impact on the risk of infection in our sample, contrary to evidence in the published literature [5, 7–9, 12]. More in line with the existing literature is the lack of an association with other social activities, such as shopping [4, 5, 7–9, 12], attending indoor sports activities [5, 7, 8, 12], and using public transport [4, 5, 7]. Overall these data seem to indicate a minimal risk when visiting places that implement strict preventive measures such as mandatory use of face masks and physical distancing, reduction of maximum capacity, and regulation of access by means of the EU digital COVID certificate [4, 8, 12, 15, 17, 19].

Notably, this is the first study in which international students were found to be more likely to become infected with SARS-CoV-2 than their peers. A case-control study found that non-native speakers had higher rates of infection, probably due to limited access to information and their tendency to self-isolate from other communities [6]. A similar explanation could apply to our foreign students, reflecting a lower propensity to adhere more closely to institution rules, and highlighting a gap in knowledge of campus preventive measures. For this reason, correctly and continuously informing international students about prevention measures to increase their compliance and consequently the effectiveness of the measures is warranted.

This study has several strengths and limitations. To the best of our knowledge, this is the first study to investigate behavioral risk factors for SARS-CoV-2 infection in a university population after anti-COVID-19 vaccination. Furthermore, by adopting a test-negative study design, we were able to exclude asymptomatic infections among controls, which would have biased the association of interest. The enrollment of incident cases that were subsequently matched to controls on a calendar basis allowed both groups to be exposed to the same preventive measures of COVID-19 mitigation. In contrast, potential information biases, such as social desirability and recall bias, are present in this study. Since the interviews were conducted after the test results were known, the results may have influenced students' responses; however, the student response rate was high. In addition, since we were only able to adjust our models for a few variables, residual confounding cannot be excluded. Finally, the study was conducted in a restricted setting and included the university population of Sapienza University of Rome. However, it is the largest university in Europe and has 122,000 students enrolled, 10% of

whom are international students. This condition allows for good variability in the sample. In addition, the opt-in procedure for the testing program may mean our students were unrepresentative of the general Sapienza University population. However, the study was conducted at a time when all degree programs had returned to in-person activities, including international students.

## Conclusions

In conclusion, this study provides a meaningful advance in the debate on whether and how to fully reopen face-to-face university campus activities. Our data clearly indicate that if universities are organized and follow strict guidelines on infection prevention measures, as was the case for Sapienza University, it is safe to attend campus activities even during an infectious disease epidemic.

The study also highlighted a few implications for practice to support the development of health policies related to the closure of HEIs: educational communities should implement communication campaigns to promote adherence to basic rules that improve collective safety to ensure that students adopt less risky behaviors both inside and outside the university. In addition, universities should remain open or reopen as soon as possible because of the potential benefits of in-person educational models on students' academic, social, mental, and physical outcomes.

Concerning the implications for research, analytical epidemiologic studies should be put in place as part of routine surveillance activities to identify risk factors for infection and also to monitor the effectiveness of the strategies adopted. Such measures should help to preserve the vital education and research missions of universities worldwide.

## Supporting information

**S1 Checklist. STROBE statement—Checklist of items that should be included in reports of** *case-control studies.*
(DOC)

## Acknowledgments

*Collaborating Group*: Pierluigi Donia[1], Barbara Tesi[1], Davide Acco[1], Alessia Crielesi[1], Valentin Imeshtari[1], Silvia Sangiorgio[3], Dino De Biase[3], Maria Antonella Zingaropoli[6], Donatella Guarino[6]

[1]Department of Public Health and Infectious Diseases, Sapienza University of Rome, Rome, Italy.

[3] Special office for Prevention, Protection and H&S Overall Supervisory, Sapienza University of Rome, Rome, Italy

[6] Microbiology and Virology Unit of the University Hospital "Policlinico Umberto I", Rome, Italy.

Contact email address [lead author]: Pierluigi Donia, pierluigi.donia@uniroma1.it.

## Author Contributions

**Conceptualization:** Erika Renzi, Giuseppe Migliara, Antonella Polimeni, Paolo Villari.

**Data curation:** Erika Renzi, Valentina Baccolini, Antonio Covelli, Leonardo Maria Siena, Antonio Sciurti, Giuseppe Migliara, Azzurra Massimi.

**Formal analysis:** Erika Renzi, Valentina Baccolini, Leonardo Maria Siena, Antonio Sciurti, Giuseppe Migliara.

**Investigation:** Erika Renzi, Antonio Covelli, Azzurra Massimi, Antonio Angeloni, Ombretta Turriziani, Guido Antonelli.

**Methodology:** Erika Renzi, Valentina Baccolini, Giuseppe Migliara, Corrado De Vito, Paolo Villari.

**Project administration:** Erika Renzi, Carolina Marzuillo, Corrado De Vito, Ombretta Turriziani, Guido Antonelli, Fabrizio D'Alba, Antonella Polimeni, Paolo Villari.

**Resources:** Leandro Casini, Antonio Angeloni, Ombretta Turriziani, Guido Antonelli, Fabrizio D'Alba, Antonella Polimeni, Paolo Villari.

**Software:** Valentina Baccolini, Leonardo Maria Siena, Antonio Sciurti, Giuseppe Migliara.

**Supervision:** Valentina Baccolini, Carolina Marzuillo, Corrado De Vito, Leandro Casini, Guido Antonelli, Fabrizio D'Alba, Antonella Polimeni, Paolo Villari.

**Validation:** Carolina Marzuillo, Corrado De Vito, Leandro Casini, Antonio Angeloni, Ombretta Turriziani, Guido Antonelli, Fabrizio D'Alba, Antonella Polimeni, Paolo Villari.

**Visualization:** Corrado De Vito, Antonella Polimeni, Paolo Villari.

**Writing – original draft:** Erika Renzi, Valentina Baccolini.

**Writing – review & editing:** Carolina Marzuillo, Corrado De Vito, Leandro Casini, Antonella Polimeni, Paolo Villari.

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
