## [Decision Letter · Decision Letter 0]

23 Jan 2024

PONE-D-23-36324Keeping university open did not increase the risk of SARS-CoV-2 acquisition: a test negative case-control study among students.PLOS ONE

Dear Dr. Renzi,

Thank you for submitting your manuscript to PLOS ONE. After careful consideration, we feel that it has merit but does not fully meet PLOS ONE’s publication criteria as it currently stands. Therefore, we invite you to submit a revised version of the manuscript that addresses the points raised during the review process.

We look forward to receiving your revised manuscript.

Kind regards,

Massimiliano Papi

Academic Editor

PLOS ONE

Journal Requirements:

-  https://doi.org/10.3389/fpubh.2022.1010130

In your revision ensure you cite all your sources (including your own works), and quote or rephrase any duplicated text outside the methods section. Further consideration is dependent on these concerns being addressed.

4. One of the noted authors is a group or consortium [Collaborating Group]. In addition to naming the author group, please list the individual authors and affiliations within this group in the acknowledgments section of your manuscript. Please also indicate clearly a lead author for this group along with a contact email address.

Reviewers' comments:

Reviewer's Responses to Questions

**Comments to the Author**

1. Is the manuscript technically sound, and do the data support the conclusions?

Reviewer #1: Yes

Reviewer #2: Yes

2. Has the statistical analysis been performed appropriately and rigorously? 

Reviewer #1: Yes

Reviewer #2: Yes

3. Have the authors made all data underlying the findings in their manuscript fully available?

Reviewer #1: Yes

Reviewer #2: Yes

4. Is the manuscript presented in an intelligible fashion and written in standard English?

Reviewer #1: Yes

Reviewer #2: Yes

5. Review Comments to the Author

Reviewer #1: This study is good for following strict scientific rules, however, in the statistical analysis section the author has not explained how to collapse scores for variables measured using a Likert scale. Explain why it is important to conduct a study before a vaccination programme is conducted.

Reviewer #2: Introduction

1. You mentioned the scarcity of analytical epidemiological studies investigating SARS-CoV-2 acquisition risk factors, particularly in healthy individuals in community settings like schools and higher education institutions (HEIs). This scarcity could affect the breadth of available data and, consequently, the depth of analysis. Was this the case in your study?

2. While the study was conducted after the release of anti-COVID-19 vaccines, it doesn't explicitly address how this timing might affect the identification of risk factors. Changes in vaccination rates, behaviors, or virus dynamics during the vaccination campaign might impact the study's findings but are not explicitly discussed.

3. The focus on Sapienza University of Rome might limit application of your study findings, as different universities might have different policies, demographics, or geographical contexts that affect the transmission dynamics of SARS-CoV-2. How was this limitation addressed in your study.

Methods

1. The method of selecting controls, contacting ten potential controls until two agreed to participate, was a potential source for introducing selection bias. Those who agreed might have differed in characteristics or behaviors from those who declined, impacting the representativeness of the control group. Did you put measures to overcome such potential bias?

2. While consent and anonymity were ensured, ethical considerations regarding data collection, especially sensitive information, might have implications that could impact the study's validity or participants' willingness to disclose certain details. How did you handle it?

Results

1. How comes in table 2 you’ve two AoR values for your first variable, gender?

Conclusion

Rewrite, what you currently have is more of a summary and not conclusion.

Introduce recommendation section.

What are the implications of your findings? Introduce a section on implications.

6. PLOS authors have the option to publish the peer review history of their article (what does this mean?). If published, this will include your full peer review and any attached files.

Reviewer #1: No

Reviewer #2: No

---

## [Author Response · Author response to Decision Letter 0]

8 Mar 2024

8 March 2024

Dear Editor,

 We are submitting the revision of the manuscript no. PONE-D-23-36324R1 entitled “Keeping university open did not increase the risk of SARS-CoV-2 acquisition: a test negative case-control study among students” that we submitted for publication on PLOS ONE.

 We made a few changes in the Methods, Discussion and Conclusions, as per the Reviewer’s comments. All suggestions made by the Reviewer have been taken into account. The following are our replies. 

Reviewer #1

Dear Reviewer,

thank you for your comments and suggestions, which were very helpful in revising the manuscript.

Below are the changes made according to your comments.

1. This study is good for following strict scientific rules, however, in the statistical analysis section the author has not explained how to collapse scores for variables measured using a Likert scale. 

1. We thank the Reviewer for this comment. We have included in the Methods, paragraph "Statistical analysis" information about the dichotomization process of the variables collected through a Likert Scale. The changes made are the following: “and exposure activity responses during the two weeks before testing for SARS-CoV-2, collected using Likert scales, were dichotomized as never vs. once or more.” (page 6, line 141).

2. Explain why it is important to conduct a study before a vaccination programme is conducted.

2. We thank the Reviewer for this comment. This case-control study was conducted following the nationwide release of the anti-COVID-19 vaccination and is aimed to identify risk factors for COVID-19 acquisition in Sapienza students after the implementation of the vaccination programme. Exploring the risk factors in the student population during the vaccination campaign is important in order to analyze the impact of vaccination on student behaviour regarding adherence to COVID-19 preventive measures (mask use, hand washing, physical distancing) and also to understand the main differences between the groups of students who accepted the vaccination or not. However, conducting case-control studies prior to the vaccination campaign is a key element in understanding the role of university and school closures during the early stages of the pandemic. For this reason, we conducted two case-control studies to identify risk factors for COVID-19 acquisition in Sapienza students before the release of the COVID-19 vaccines and produced consistent results in this new case-control study. Below you may find references to the case-control studies conducted by Sapienza before the vaccine programme:

- Migliara G, Renzi E, Baccolini V, Cerri A, Donia P, Massimi A, Marzuillo C, De Vito C, Casini L, Polimeni A, Gaudio E, Villari P, The Collaborating Group. Predictors of SARS-CoV-2 Infection in University Students: A Case-Control Study. Int J Environ Res Public Health. 2022 Nov 3;19(21):14376. doi: 10.3390/ijerph192114376. PMID: 36361257; PMCID: PMC9655450.

- Baccolini V, Siena LM, Renzi E, Migliara G, Colaprico C, Romano A, Massimi A, Marzuillo C, De Vito C, Casini L, Antonelli G, Turriziani O, Angeloni A, D'Alba F, Villari P, Polimeni A; Collaborating Group. Prevalence of SARS-CoV-2 infection and associated risk factors: A testing program and nested case-control study conducted at Sapienza University of Rome between March and June 2021. Front Public Health. 2022 Oct 19;10:1010130. doi: 10.3389/fpubh.2022.1010130. PMID: 36339150; PMCID: PMC9627192.

8 March 2024

Dear Editor,

 We are submitting the revision of the manuscript no. PONE-D-23-36324R1 entitled “Keeping university open did not increase the risk of SARS-CoV-2 acquisition: a test negative case-control study among students” that we submitted for publication on PLOS ONE.

 We made a few changes in the Methods, Discussion and Conclusions, as per the Reviewer’s comments. All suggestions made by the Reviewer have been taken into account. The following are our replies. 

Reviewer #2

Dear Reviewer,

thank you for your comments and suggestions, which were very helpful in revising the manuscript.

Below are the changes made according to your comments.

Introduction

1. You mentioned the scarcity of analytical epidemiological studies investigating SARS-CoV-2 acquisition risk factors, particularly in healthy individuals in community settings like schools and higher education institutions (HEIs). This scarcity could affect the breadth of available data and, consequently, the depth of analysis. Was this the case in your study?

1. We thank the Reviewer for this comment. The scarcity of data from analytical epidemiology studies concerning the identification of risk factors for SARS-CoV-2 infection had no influence on the data collection of our study. An analytical methodological epidemiology approach was adopted for the identification of risk factors, data collection was based on a self-structured questionnaire based on the risk factors identified by WHO and CDC for COVID-19. Furthermore, it was adapted to the national context and the health policies implemented by the government. The use of analytical study methodology and the related adjustment of the survey to the national context should have helped to preserve the depth of the analysis.

2. While the study was conducted after the release of anti-COVID-19 vaccines, it doesn't explicitly address how this timing might affect the identification of risk factors. Changes in vaccination rates, behaviors, or virus dynamics during the vaccination campaign might impact the study's findings but are not explicitly discussed.

2. We thank the Reviewer for this comment. We agree changes in students’ behavior or virus dynamics during the vaccination campaign could have had an impact on the results of the study. Therefore, multiple aspects that could be affected by the release of anti-COVID-19 vaccination, such as adherence to preventive measures, or COVID-19 vaccination status, were considered and investigated, at first in univariable analyses and then in multivariable analysis, but no significant difference was found between cases and controls as for changes in behavior following the vaccination release, while a higher likelihood of SARS-CoV-2 acquisition was found according to the vaccination status, as also confirmed by the existing scientific literature. However, to better discuss this finding, we expanded the Discussion section as follows: “Moreover, students who had received only the primary course of COVID-19 vaccine were more at risk of infection with SARS-CoV-2 than those who had received their booster dose. The efficacy of two doses of COVID-19 vaccine in preventing infection and symptomatic disease decreases by 20-30% six months after vaccination [49] and the booster dose, as described in recent studies, seems to increase protection against the Omicron variant [50]”(page 14 lines 331 -336).

3. The focus on Sapienza University of Rome might limit application of your study findings, as different universities might have different policies, demographics, or geographical contexts that affect the transmission dynamics of SARS-CoV-2. How was this limitation addressed in your study.

3. We thank the Reviewer for this comment. We revised the Discussion section adding the following sentence in the limitation paragraph: “Finally, the study was conducted in a restricted setting and included the university population of Sapienza University of Rome. However, it is the largest university in Europe and has 122,000 students enrolled, 10% of whom are international students. This condition allows for good variability in the sample” (page 16 lines 366 - 369).

Methods

1. The method of selecting controls, contacting ten potential controls until two agreed to participate, was

a potential source for introducing selection bias. Those who agreed might have differed in characteristics

or behaviors from those who declined, impacting the representativeness of the control group. Did you put measures to overcome such potential bias?

1. We thank the Reviewer for this comment. The control selection process involved a random drawing of 10 names of students who were swab negative for SARS-CoV-2 on the same day as the cases. These negative students were contacted by telephone in order of extraction by the Department of Public Health, which provided the general information of the study objectives; after the controls viewed the study aims, they could decide whether to participate. As reported in the methods, the percentage of students who refused to participate in the study was 16 (6% of all controls interviewed). Therefore, 244 controls were interviewed, of which only 16 were selected beyond the first two names on the list (page 8 Lines 200 - 202). A small percentage of controls that did not give consent allowed us to reduce any potential selection bias.

2. While consent and anonymity were ensured, ethical considerations regarding data collection, especially sensitive information, might have implications that could impact the study's validity or participants' willingness to disclose certain details. How did you handle it?

2. We thank the Reviewer for this comment. The following security measures were taken to protect personal data: the student was randomly assigned an alphanumeric code, and following consent to participate in the study, all directly identifiable data (first name, last name, mobile phone, and e-mail) were removed from the dataset. The student's personal data were recorded under an alphanumeric code that was transcribed by the student, who was the only one who could associate the interview data with directly identifiable personal data. Sensitive data (COVID-19 positivity data, anti-COVID-19 vaccination, presence of chronic conditions) held by the researchers were not associated with a directly identifiable student. The procedure was explained to students at the stage of acquiring consent to participate in the research study, and formal consent to the processing of personal data was provided. This procedure ensured that Sapienza University of Rome could not associate the information provided with the student's identity, reducing possible bias.

Results

1. How comes in table 2 you’ve two AoR values for your first variable, gender?

1. We regret any lack of clarity in Table 2. The unadjusted OR shown under the variable "Gender" is related to the variable age, expressed in years. It was included in the variable description row (Age, years) because it was a continuous variable without categorical baseline information. 

Unadjusted OR for Gender = 0.99 (0.60-1.60)

Unadjusted OR for Age, years = 1.01 (0.95-1.07)

Conclusion

1. Rewrite, what you currently have is more of a summary and not conclusion.

Introduce recommendation section. What are the implications of your findings? Introduce a section on implications.

1. We thank the Reviewer for this suggestion. We revised the Conclusion section adding details concerning the implication for practice and research. As suggested, we reduced the summary section. We changed the text as follows: “In conclusion, this study provides a meaningful advance in the debate on whether and how to fully reopen face-to-face university campus activities. Our data clearly indicate that if universities are organized and follow strict guidelines on infection prevention measures, as was the case for Sapienza University, it is safe to attend campus activities even during an infectious disease epidemic. The study also highlighted a few implications for practice to support the development of health policies related to the closure of HEIs: educational communities should implement communication campaigns to promote adherence to basic rules that improve collective safety to ensure that students adopt less risky behaviors both inside and outside the university. In addition, universities should remain open or reopen as soon as possible because of the potential benefits of in-person educational models on students' academic, social, mental, and physical outcomes. Concerning the implications for research, analytical epidemiologic studies should be put in place as part of routine surveillance activities to identify risk factors for infection and also to monitor the effectiveness of the strategies adopted. Such measures should help to preserve the vital education and research missions of universities worldwide.” (page 16 Lines 375 - 390).

8 March 2024

Dear Editor,

 We are submitting the revision of the manuscript no. PONE-D-23-36324R1 entitled “Keeping university open did not increase the risk of SARS-CoV-2 acquisition: a test negative case-control study among students” that we submitted for publication on PLOS ONE.

All suggestions made by the Journal Requirements have been considered. The following are our replies. 

Journal Requirements:

1. Thank you for the suggestion. We checked the editorial guidelines.

- https://doi.org/10.3389/fpubh.2022.1010130

In your revision ensure you cite all your sources (including your own works), and quote or rephrase any duplicated text outside the methods section. Further consideration is dependent on these concerns being addressed.

2. Thanks for the suggestion. We assessed the presence of some minor overlapping regarding https://doi.org/10.3389/fpubh.2022.1010130. The overlap appears in the Materials and Methods section, where the article https://doi.org/10.3389/fpubh.2022.1010130 is reported as it represented the first case-control study conducted by our research group (same authors) using the same methodology but on a sample of students not yet vaccinated. The present study is a second investigation carried out at different times to assess any changes in risk factors for acquiring SARS-CoV-2 after receiving the anti-COVID-19 vaccination. This overlap relates specifically to the setting (testing program) and the choice of using a conditional logistic regression model that are common to both studies. However, the contents in common with the studies have been reformulated.

3.Thank you for the suggestion.

4. One of the noted authors is a group or consortium [Collaborating Group]. In addition to naming the author group, please list the individual authors and affiliations within this group in the acknowledgments section of your manuscript. Please also indicate clearly a lead author for this group along with a contact email address.

4. Thank you for the suggestion. We have listed the names, affiliations, and a reference e-mail contact for the collaborating group in the Acknowledgments section (page 17 lines 401 - 408).

5. Please review your reference list to ensure that it is complete and correct. If you have cited papers that have been retracted, please include the rationale for doing so in the manuscript text or remove these references and replace them with relevant current references. Any changes to the reference list should be mentioned in the rebuttal letter that accompanies your revised manuscript. If you need to cite a retracted article, indicate the article’s retracted status in the References list and also include a citation and full reference for the retraction notice.

5. Thank you for the suggestion. Thank you for the suggestion. We have checked the bibliography and updated citation no. 7, including the erratum reference (page 19 lines 473 - 486).

Fisher KA, Tenforde MW, Feldstein LR, Lindsell CJ, Shapiro NI, Files DC, Gibbs KW, Erickson HL, Prekker ME, Steingrub JS, Exline MC, Henning DJ, Wilson JG, Brown SM, Peltan ID, Rice TW, Hager DN, Ginde AA, Talbot HK, Casey JD, Grijalva CG, Flannery B, Patel MM, Self WH; IVY Network Investigators; CDC COVID-19 Response Team. Community and Close Contact Exposures Associated with COVID-19 Among Symptomatic Adults ≥18 Years in 11 Outpatient Health Care Facilities - United States, July 2020. MMWR Morb Mortal Wkly Rep. 2020 Sep 11;69(36):1258-1264. doi: 10.15585/mmwr.mm6936a5. Erratum in: MMWR Morb Mortal Wkly Rep. 2020 Sep 25;69(38):1380. PMID: 32915165; PMCID: PMC7499837.

---

## [Editor Report · Decision Letter 1]

12 Mar 2024

Keeping university open did not increase the risk of SARS-CoV-2 acquisition: a test negative case-control study among students.

PONE-D-23-36324R1

Dear Dr. Erika Renzi,

We’re pleased to inform you that your manuscript has been judged scientifically suitable for publication and will be formally accepted for publication once it meets all outstanding technical requirements.

Kind regards,

Massimiliano Papi

Academic Editor

PLOS ONE
---

## [Editor Report · Acceptance letter]

18 Mar 2024

PONE-D-23-36324R1 

PLOS ONE

Dear Dr. Renzi, 

I'm pleased to inform you that your manuscript has been deemed suitable for publication in PLOS ONE. Congratulations! Your manuscript is now being handed over to our production team.

Kind regards, 

on behalf of

Prof. Massimiliano Papi 

Academic Editor

PLOS ONE